# Reporting quality in preclinical animal experimental research in 2009 and 2018: A nationwide systematic investigation

**Birgitte S. Kousholt**[1]*, **Kirstine F. Præstegaard**[1], **Jennifer C. Stone**[2], **Anders Fick Thomsen**[1], **Thea Thougaard Johansen**[1], **Merel Ritskes-Hoitinga**[1,3], **Gregers Wegener**[1,4]

**1** Department of Clinical Medicine, AUGUST, Aarhus University, Aarhus, Denmark, **2** Department of Health Services Research and Policy, Research School of Population Health, Australian National University, Canberra, Australia, **3** Faculty of Medical Sciences, Radboud University, Nijmegen, The Netherlands, **4** Department of Clinical Medicine, Translational Neuropsychiatry Unit, Aarhus University, Aarhus, Denmark

* birgitte.kousholt@clin.au.dk

**Data Availability Statement:** All data files are available from the Open Science Framework database: https://osf.io/7e829/.

## Abstract

Lack of translation and irreproducibility challenge preclinical animal research. Insufficient reporting methodologies to safeguard study quality is part of the reason. This nationwide study investigates the reporting prevalence of these methodologies and scrutinizes the reported information's level of detail. Publications were from two time periods to convey any reporting progress and had at least one author affiliated to a Danish University. We retrieved all relevant animal experimental studies using a predefined research protocol and a systematic search. A random sampling of 250 studies from 2009 and 2018 led to 500 publications in total. Reporting of measures known to impact study results estimates were assessed. Part I discloses a simplified two-level scoring "yes/no" to identify the presence of reporting. Part II demonstrates an additional three-level scoring to analyze the reported information's level of detail. Overall reporting prevalence is low, although minor improvements are noted. Reporting of randomization increased from 24.0% in 2009 to 40.8% in 2018, blinded experiment conduct from 2.4% to 4.4%, blinded outcome assessment from 23.6% to 38.0%, and sample size calculation from 3.2% to 14.0%. Poor reporting of details is striking with reporting of the random allocation method to groups being only 1.2% in 2009 and 6.0% in 2018. Reporting of sample size calculation method was 2.4% in 2009 and 7.6% in 2018. Only conflict-of-interest statements reporting increased from 37.6% in 2009 to 90.4%. Measures safeguarding study quality are poorly reported in publications affiliated with Danish research institutions. Only a modest improvement was noted during the period 2009–2018, and the lack of details urgently prompts institutional strategies to accelerate this. We suggest thorough teaching in designing, conducting and reporting animal studies. Education in systematic review methodology should be implemented in this training and will increase motivation and behavior working towards quality improvements in science.

**Funding:** BSK received funding from the Danish 3R Center, grant number 33010-NIFA-19-705, https://en.3rcenter.dk/ and Ester M. og Konrad Kristian Siggurdssons Dyrevaernsfond, https://www.dyrevaernsfond.com/ The funders had no role in study design, data collection and analysis, decision to publish, or preparation of the manuscript.

**Competing interests:** The authors have declared that no competing interests exist.

## Introduction

Poor reproducibility and translational failure in biomedical research lead to skepticism regarding the reliability of preclinical research findings. The reasons are multi-factorial [1–4]. A prevalent issue is unsatisfactory internal validity. Internal validity is the extent to which a design and conduct of a study eliminates the possibility of systematic errors (bias) [4]. Appropriate methodologies safeguarding against systematic errors can be implemented in the design, conduct, and analysis of an experiment in order to increase the internal validity [4].

Essential safeguards are blinding, randomization, and a thorough description of animals' and samples' flow including reasons for exclusion [5]. The judgment of the scientific evidence is hampered if these measures are poorly reported [6]. Evidence exists that lack of reporting corresponds to the absence of conduct [7, 8]. Systematic reviews of preclinical animal studies disclose smaller effect sizes when randomization and blinding are implemented compared with studies not reporting these precautions [9–12]. This finding is corroborated in meta-epidemiological studies of clinical data that identify a negative additive impact when more than one safeguard is omitted [13–16]. Attrition bias (i.e., poor handling of dropouts) skews data and jeopardizes a study's scientific robustness. Holman et al. demonstrate that losses of only a few animals in a study can distort study effects [17]. A safeguard of importance is how the sample size is reached. Preclinical animal experiments often carry (too) small group sizes. A drawback to this is that positive findings may be due to chance rather than actual effect [18–20]. Thus, comprehensive sample size calculations based on the best available evidence are paramount. Other influential quality factors are, for example, the animals' health status or comorbidities before and during experiments, as undetected diseases may affect the study outcome [21–23].

The Animal Research: Reporting of In Vivo Experiments (ARRIVE) guidelines for animal experiments reporting, first published in 2010, provide recommendations on improving low reporting standards and have recently been updated [24, 25]. However, studies repeatedly show inadequate reporting of quality indicators [26–30], suggesting the unsuccessful implementation of the guidelines, even though over 1000 journals endorse them. The implementation may be hindered by the lack of engagement of multiple stakeholders who all must engage in improving the reporting quality. In this context, the use of the ARRIVE guideline by researchers is necessary already at the planning stage to help improve experimental design and, in turn, improve reporting. Previous research has investigated the prevalence of reporting of measures to reduce the risk of bias for specific animal disease models or subjects of interest [28–31]. Other previous evaluations of preclinical reporting have provided an overview of the reporting status of items related to the internal validity or rigor of these experiments (e.g. blinding and randomization) [32]. This study investigates the reported information's level of detail by assessing preclinical studies within all animal experimental research fields with one or more authors affiliated with Danish research institutions. In part I of the study, we focus on the overall reporting status of methodological safeguards. In part II, the focus is on the level of detail given for each reported item. To detect whether progress over the years exists, we investigated publications containing experiments published before (the year 2009) and after (the year 2018) publication of the ARRIVE guidelines [24].

## Materials and methods

The experimental design was based on random sampling to avoid bias. An equal number of studies from each year were included to compare the results between the two time periods. It was estimated that a thorough assessment of 500 papers in total– 250 papers from each year– could be performed within the given timeframe.

## Study protocol

To further prevent methodological flaws and minimize bias, we modified a pre-specified systematic review protocol for animal intervention studies offered by the SYstematic Review Centre for Laboratory animal Experimentation (SYRCLE) [33]. The protocol was uploaded at SYRCLE (7th of November 2018) for guidance and feedback and is found in the supporting information (S1 File).

## Selection of studies

In collaboration with a library information specialist, we retrieved all potentially relevant studies using a modified, comprehensive search strategy [34, 35]. The search was systematically performed in two databases, Medline (via PubMed) and Embase. All *in vivo* studies conducted in non-human vertebrates with one or more authors affiliated with at least one of five Danish universities of interest were retrieved. The search was divided into two separate searches based on publication year (search 1: 1st of January 2018 until 6th November 2018; search 2: the year 2009). The studies were imported to two dedicated EndNote libraries (EndnoteX8, Clarivate Analytics, Philadelphia, USA), and duplicates were removed. One thousand, one hundred and sixty-one studies from 2009 and 1890 studies from 2018 were found.

The information from the Endnote libraries was copied to Excel (MS Office, version 2016, Microsoft Corp., USA), and the publications from each year were randomized using the "= RAND()" command, thereby allocating a unique random number to each publication. Due to the decision to perform a comprehensive search strategy to identify all relevant preclinical animal studies, the majority of the studies were not applicable. To meet the goal of including the 250 relevant publications from each year, publications were imported consecutively in the randomized order (first 500 studies each from the year 2009 and 2018, then 250 studies and lastly 150 studies from each year) into a systematic review manager software program, Covidence (Covidence, Melbourne, Australia) [36]. A total of 1800 studies (out of 3051) were screened for eligibility. Two hundred and fifty-six from 2009 and 275 from 2018 were found eligible. Of these, 250 publications from each year were selected based on the random sampling allocation sequence. The exclusion of studies was based on the following exclusion criteria: science related to farming, wild animals or invertebrates, environment, human (clinical) studies, in vitro research, not primary papers/publications, lack of abstract or full text, studies containing no intervention or no Danish author affiliation, and exploratory studies (the latter studies were identified through study author statements that the study was explorative, or studies were assessed to investigate novel questions and to be hypothesis-generating). Further information on the distribution of excluded studies is found in supporting information (S2 File). The flow diagram for random sampling, screening, and selection of publications is shown in Fig 1.

## Data extraction and analysis

The Covidence Risk of Bias (RoB) tool was selectively modified for this study's aims and to assess reporting quality in compliance with SYRCLE's RoB tool for animal studies [37]. Each publication was assessed according to 10 items primarily based on the Landis four related to the quality of reporting of significant methodology and included in the ARRIVE guidelines [5, 24, 25]. The selection of items was due to the nature of the study capturing different types of animal research. One item "health status", was chosen since it, to our knowledge, is scarcely investigated even though it may influence many research outcomes [21]. The reporting quality form and algorithms for scoring are included in the supporting information (S1 Table). Two independent reviewers (KFP and JCS) assessed publications for reporting quality, each blinded to the other's assessment. Reviewers examined the full text of the articles, including figures and

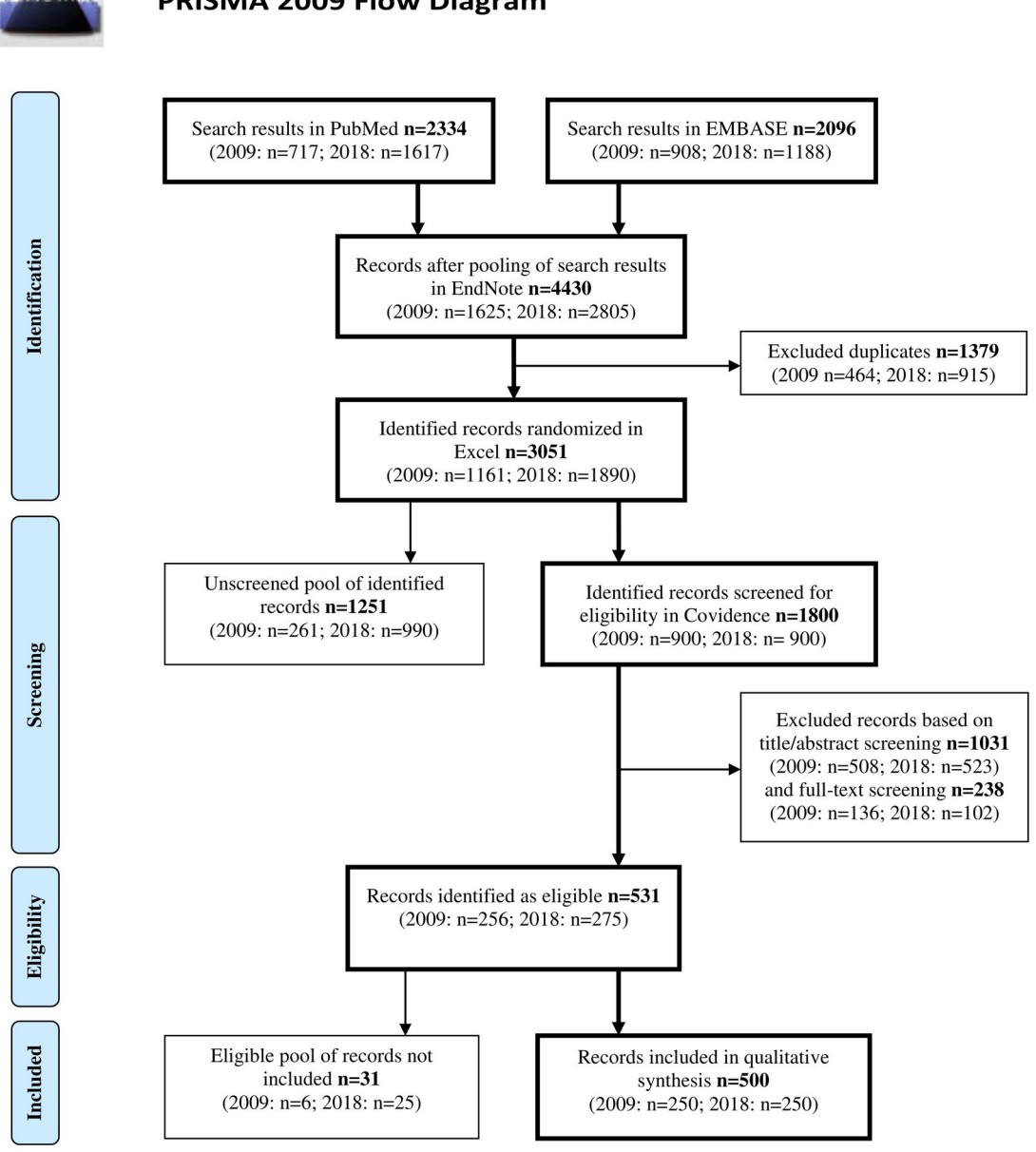

**Fig 1. Flow diagram describing the random sampling, screening, and selection of studies.**

tables, and supplemental information, but references to other studies were not evaluated. Our approach for assessing the reporting quality included three steps:

**Step 1**: To investigate the overall reporting status (Part I) of the selected items, each item was operationalized such that we scored a result of "Yes" or "No" in Covidence. Publications were qualitatively scored "Yes" if the specific item was reported or "No" when there was no reporting of the item or when criteria for "Yes" were not met. "Unclear" or "Partial" scores were not used. In instances where the item was only partially reported and did not contain the

complete information defined in the item or where items were reported not conducted (e.g., authors reported that randomization was not conducted), the study was scored as "Yes" and notes provided. Details of this process are given in the supporting information (S1 Table). Each item's annotations and quotes were selected and saved for subsequent data quantification. This extra step made judgment decisions during this review consistent.

**Step 2**: After completing each reviewer's initial reporting quality assessments, a consensus of the reporting quality results was undertaken in Covidence. If both reviewers agreed on the item, the final judgment defaulted to the agreed value leaving discrepant items for further assessment. Discrepancies were resolved, and the consensus was reached through discussion and the inclusion of a third reviewer (BSK).

**Step 3**: After completing the assessment in Covidence, data were extracted and sorted in MS Excel. After that, a numerical score of 1, 2, 3, or 0—where 0 corresponds to no information—was given according to the quality of information (quotes and comments) saved for each item described in step 1 (Part II). Details of this process are provided in the supporting information S1 Table, and our criteria for aiding judgment and associated examples of quotes are found in the supporting information (S2 Table).

Survey data were analyzed using MS Excel and Stata Statistical Software: Release 16.1 (Stata Corp. 2019. College Station, TX: StataCorp LLC). Descriptive statistics were generated for all items and were presented in bar graphs and tables. Prevalence and differences between prevalence for the 2009 and 2018 studies were reported with 95% confidence intervals. Reviewer agreement and Cohen's Kappa values are disclosed in the supporting information (S3 File).

## Results

The flow of the publications retrieved is described in the Methods section and in Fig 1. Five hundred publications were included in the investigation, 250 from 2009 and 250 from 2018 according to the procedure described in the Methods section. A simplified two-level scoring (reported "yes" or not reported "no") is given in part I. Part II discloses the results of a three-level scoring (1, 2, and 3) system where one comprised the least detailed information conveyed. The results of part I are presented graphically in Fig 2 and the results of part II are shown in Table 1.

### Part I: The overall reporting status

Approximately half of the reviewed publications from 2009 reported a study sample size given as a number (53.2%, CI 46.8–59.5%) compared to 152 publications (60.8%, CI 54.5–66.9%) in 2018. Eight publications (3.2%, CI 1.4–6.2%) from 2009 reported on sample size calculation as a measure of reassurance that studies were adequately powered. This number increased to 35 publications (14.0%, CI 10.0–18.9%) in 2018 (Fig 2A). Random allocation of animals to experimental groups was reported in 60 publications (24.0%, CI 18.8–29.8%) from 2009 (out of 234 in which this would have been appropriate; 93.6%), which increased to 102 publications (40.8%, CI 34.7–47.2%) in 2018 (out of 233 in which this would have been appropriate; 93.2%). Blinded experiment conduct was reported in six publications (2.4%, CI 0.9–5.2%) from 2009 and increased to 11 in 2018 (4.4%, CI 2.2–7.7%), and blinded outcome assessment was reported in 59 publications (23.6%, CI 18.5–29.4%) from 2009 and increased to 95 (38.0%, CI 32.0–44.3%) in 2018 (Fig 2B).

Information regarding the numbers of samples or animals in the result section (attrition I) was reported in 195 publications (78.0%, CI 72.4–83.0%) from 2009 and increased to 206 (82.4%, CI 77.1–86.9%) in 2018. Consistency in the numbers of samples or animals between

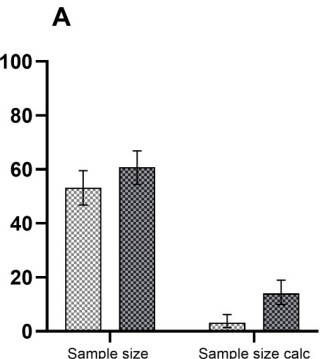

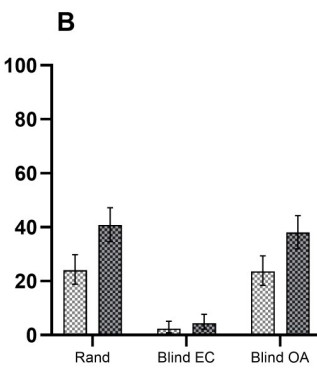

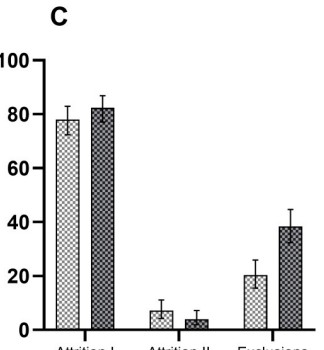

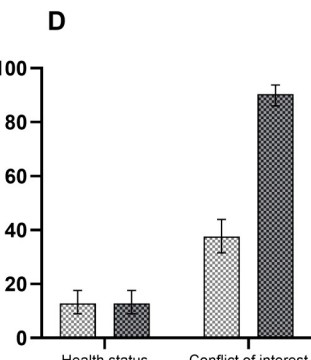

**Fig 2. Prevalence of reporting quality in Danish preclinical research in 2009 compared to 2018.** *Left Y-axis*: prevalence of reporting in %. *X-axis*: (**A**) sample size and sample size calculation, (**B**) randomization (Rand), blinded experiment conduct (Blind EC), and blinded outcome assessment (Blind OA), (**C**) attrition I (reporting numbers of samples/animals in the result section), attrition II (reporting consistency in numbers of samples/animals between methods and result section), and exclusions (reporting numbers of samples/animals excluded), (**D**) health status and conflict of interest. The error bars represent 95% confidence intervals. Light grey bars: 2009. Dark grey bars: 2018.

the methods and the result section (attrition II) was reported in 18 publications (7.2%, CI 4.3–11.1%) in 2009. This number decreased to 10 publications (4.0%, CI 1.9–7.2%) in 2018.

Exclusions of samples or animals were reported in 51 publications (20.4%, CI 15.6–25.9%) from 2009. In 2018, this number was 96 (38.4%, CI 32.3–44.7%) (Fig 2C). The animal health status was reported in 32 of the publications surveyed (12.8%, CI 8.9–17.6%) in 2009, and the number did not change in 2018 (12.8%, CI 8.9–17.6%). The reporting on conflicts of interest increased from 94 publications (37.6%, CI 31.6–43.9%) in 2009 to 226 publications (90.4%, CI 86.1–93.8%) in 2018 (Fig 2D).

Table 1. Prevalence of reporting details in Danish preclinical research in 2009 compared to 2018.

| Item | Details of reporting | 2009 (n = 250) | | 2018 (n = 250) | | 2018-2009[a] | |
|---|---|---|---|---|---|---|---|
| | | % | 95% CI | % | 95% CI | % | 95% CI |
| Sample size calculation | Reported *not performed* | 0.4 | [0.0–2.2] | 1.2 | [0.3–3.5] | 0.8 | [-0.8–2.4] |
| | Reported performed | 0.4 | [0.0–2.2] | 5.2 | [2.8–8.7] | 4.8 | [1.9–7.7] |
| | Reported performed and calculation disclosed | 2.4 | [0.9–5.2] | 7.6 | [4.6–11.6] | 5.2 | [1.4–9.0] |
| Randomization | Reported performed | 22.8 | [17.8–28.5] | 34.8 | [28.9–41.1] | 12.0 | [4.1–19.9] |
| | Reported performed and method disclosed | 1.2 | [0.3–3.5] | 6.0 | [3.4–9.7] | 4.8 | [1.6–8.0] |
| Blinded experiment conduction | Reported *not performed* | 0.4 | [0.0–2.2] | 2.8 | [1.1–5.7] | 2.4 | [0.2–4.6] |
| | Reported performed | 2.0 | [0.7–4.6] | 1.6 | [0.4–4.1] | -0.4 | [-2.7–1.9] |
| Blinded outcome assessment | Reported *not performed* | 0.4 | [0.0–2.2] | 3.6 | [1.7–6.7] | 3.2 | [0.8–5.6] |
| | Reported performed | 23.2 | [18.1–28.9] | 34.4 | [28.5–40.7] | 11.2 | [3.3–19.1] |
| Attrition I | Reported but not for all analyses | 45.6 | [39.3–52.0] | 52.8 | [46.4–59.1] | 7.2 | [-1.5–15.9] |
| | Reported with exact numbers for all analyses | 32.4 | [26.6–38.6] | 29.6 | [24.0–35.7] | -2.8 | [-10.9–5.3] |
| Exclusions | Reported but without numbers and reason for exclusion | 6.8 | [4.0–10.7] | 16.0 | [11.7–21.1] | 9.2 | [3.7–14.7] |
| | Reported no exclusions/all included | 2.0 | [0.7–4.6] | 4.0 | [1.9–7.2] | 2.0 | [-1.0–5.0] |
| | Reported with numbers and reason for exclusion | 11.6 | [7.9–16.2] | 18.4 | [13.8–23.8] | 6.8 | [0.6–13.0] |
| Animal health status | Reported without further information | 12.8 | [8.9–17.6] | 12.4 | [8.6–17.1] | -0.4 | [-6.2–5.4] |
| | Reported and detailed information disclosed | 0.0 | [0.0–1.46] | 0.4 | [0.0–2.2] | 0.4 | [-0.4–1.2] |
| Conflict of interest | Reported with a conflict of interest present | 6.8 | [4.0–10.7] | 18.4 | [13.8–23.8] | 11.6 | [5.9–17.3] |
| | Reported with a conflict of interest absent | 30.8 | [25.1–36.9] | 72.0 | [66.0–77.5] | 41.2 | [33.2–49.2] |

CI, confidence interval; *n*, the total number of publications

[a] The relative reporting difference between 2009 and 2018

## Part II: Level of detail of reported items

Further analysis of information is presented in Table 1 and revealed that of the publications reporting a sample size calculation, six (2.4%) from 2009 provided information regarding how the sample size was chosen and described the method employed. In 2018, this number was 19 (7.6%). The remaining publications that reported on sample size calculation either did not include a calculation method (1 (0.4%) from 2009 and 13 (5.2%) from 2018) or stated that a sample size calculation was not performed (1 (0.4%) from 2009 and 3 (1.2%) from 2018).

Three publications (1.2%) from 2009 reporting random allocation also disclosed the method used. In 2018, the number was 15 publications (6.0%). Notably, 16 (6.4%) and 17 (6.8%) publications from 2009 and 2018, respectively, were either strain studies (e.g., the phenotype of transgenic animals was being compared to wild type phenotype) (2009: 15/16; 2018: 13/17) or pre-post experimental studies (2009: 1/16; 2018: 4/17) where randomizing treatment order was not feasible due to carryover effects. However, none of these publications reported that randomization was not applicable in these types of studies.

Of the publications from 2009 reporting on blinded conduct of the experiment, one publication (0.4%) reported that blinding was not conducted. This increased to seven publications (2.8%) in 2018.

Of the publications reporting on blinded outcome assessment, one publication (0.4%) from 2009 reported that blinding was not conducted. This increased to nine publications (3.6%) in 2018.

Further analysis of information regarding attrition revealed that only 81 (32.4%) of the publications reporting on numbers of samples or animals in the result section (attrition I) from 2009 reported exact numbers for all analyses. This number decreased to 74 (29.6%) in 2018.

The remaining publications (114 (45.6%) from 2009 and 132 (52.8%) from 2018) either did not report exact numbers or did not report numbers for all analyses in the study.

For details regarding exclusions of samples or animals, 29 of the publications (11.6%) in 2009 versus 46 (18.4%) in 2018 reported exact numbers and reasons for exclusion. The remaining publications reporting on exclusions either failed to report precise numbers or reasons for exclusion (17 (6.8%) in 2009 and 40 (16.0%) in 2018) or reported no exclusions or all included in the study (five (2.0%) in 2009 and 10 (4.0%) in 2018). Two publications from 2009 and one publication from 2018 provided a flow chart to clarify the animal flow.

When analyzing publications for providing detailed animal health information, one publication (0.4%) from 2018 included a health report of the experimental animals in the supplementary files.

Information regarding conflict of interest showed that 17 publications (6.8%) reported a conflict of interest and 77 publications (30.8%) reported a conflict of interest to be absent in 2009. These numbers increased in 2018 to 46 (18.4%) and 180 (72.0%), respectively.

## Discussion

This study investigated the reporting prevalence of central methodological items safeguarding study quality in experimental animal research. The aim was to get an overview of the current status and further research the detail in the information conveyed. To gain further insight into progress over time, we surveyed two time periods, 2009 and 2018. All publications had at least one researcher affiliated with a Danish research institution.

Our results from 2009 correspond well with an investigation from 2010 by Macleod et al., who surveyed preclinical research studies in 2009–2010 from leading UK universities [27]. We also found only modest improvements over time in reporting randomization, blinding, and sample size calculation, whereas the reporting of conflict of interest increased considerably. Our investigation highlights that while this topic has been extensively addressed both in the scientific community and through the development of reporting guidelines [24–31, 38, 39], reporting remains insufficient. There is still considerable room for improvement to strengthen the validity of most published pre-clinical animal studies in the light of the assumption that lack of reporting corresponds to limited conduct.

We further researched the level of detail in the information disclosed and found the level of detail was very limited. Randomization and blinding are essential methodological techniques to help reduce the influence of bias on the study outcome. Despite their importance, transparency in reporting these items was insufficient. In studies where blinding and randomization were not feasible, the reason (e.g., study design) was rarely justified nor considered a limitation and acknowledged in the study report. A description of why such a precaution is *not* taken will bring the reader's attention to the missing safeguard so the results can be judged accordingly. Many studies additionally have very complex study designs and the precautions taken to limit bias should be sufficiently reported.

In general, essential details related to randomization, such as the allocation method and sequence, were rarely conveyed. This fact is corroborated in studies of specific animal models of acute lung injury by Avey et al. They operationalized the ARRIVE guidelines to determine completeness in reporting and found no random sequence generation reporting [30]. Ting et al. similarly disclosed that no studies revealed the allocation method in experimental animal studies of rheumatology [28]. Our investigation concludes that there seems to be a general challenge across study fields.

Interestingly, small sample sizes may negatively influence successful randomization as groups may be unbalanced on critical prognostic variables. Underpowered experiments will

give less precise estimates of treatment effects. This risk can be accounted for by using appropriate methods for sample size calculation.

Only a few publications provided sufficient information regarding if and how sample size was calculated for sample sizes' exact values. In some publications, historical precedent rather than reliable statistics formed the basis for reporting the number of animals per group. We are puzzled that such unjustified scientific information is forwarded through a review process. In a study by Gulin et al., investigating compliance with ARRIVE guidelines in studies of experimental animal models for Chagas disease, there was no reporting of sample size calculation. Authors of the investigation speculated that "animal numbers were more a matter of habit than a statistical decision" [29]. This speculation highlights that results may sometimes be due to chance rather than an actual effect. A more recent study of the veterinary literature that focused on reporting adherence to the ARRIVE guidelines found missing sample size calculations to be present in both ARRIVE guideline supporting and non-supporting journals, indicating that a journal's support for ARRIVE guidelines has not to date resulted in improved reporting of these guidelines and other essential indicators of study design quality [31]. If the planned sample size is not derived statistically, this should be explicitly stated along with the rationale for the intended sample size (e.g., exploratory nature). We found information on whether a study was confirmatory or exploratory sparse. This information poses an additional problem to how much weight can be ascribed to the published results.

Systematic differences between animals completing a study and the excluded animals can introduce bias to the study results–a bias known as attrition bias [17]. Despite the importance of emphasizing and reporting exact numbers of animals at the beginning of the study and the end of the study and how many animals were excluded during the study and for which reasons, we found most studies failed to report consistently. Most publications failed to report exact numbers and reasons for exclusion, and even a decrease in reporting of animal numbers in 2018 was seen when compared with 2009. Several studies reporting on the number of samples or animals used demonstrated inconsistencies in reporting between the methods section and the results section. Only one publication from 2018 included this information objectively in a flow chart compared to two publications in 2009. A flowchart illustrating each animal's fate and the derived samples or measurements would be effective in providing the reader with a thorough overview.

An uncommonly reported item and, to our knowledge, rarely investigated item is the animals' general health status. In our study, this was one of the most poorly reported items, and only one publication from 2018 included a health report with details of the specific agents for which the animals were screened. This finding is disturbing since infections and/or comorbidities influence disease outcomes in both preclinical animal research and treatment and pathology in patients [40]. Documenting these details is essential in understanding the discrepancies seen in laboratory results [25, 41]. In our experience, many researchers do not take this fact into account. A fully disclosed health report should be mandatory and based on a case-oriented approach to the FELASA (Federation of European Laboratory Animal Science Associations) guidelines [22, 23]. Moreover, the impact of animal health on study outcomes is complex and warrants further investigation.

We envisaged that an improvement in methodological reporting would be noticeable since many journals have endorsed the ARRIVE guidelines. However, advancements continue to progress at a slow pace or do not happen at all. We show that the reported information's level of detail is generally incomplete. The incomplete reporting of these details directly impedes the ability to assess the validity of the experiments. When research cannot be assessed on its methodological rigor, it becomes less valuable and thus is a waste of essential resources and animal lives. The translation of research findings into therapeutic applications becomes highly

unreliable, and there is a high risk of guiding research in the wrong direction. Stakeholders such as funders and publishers may incite study quality, but perhaps the essential science stakeholders are researchers themselves. Researchers must conduct responsible research of high quality and have the ability to do so. This may also call for new evaluation methods [42].

Nevertheless, to conduct high-quality research, researchers need to be allocated time, understand the importance of research integrity, be trained in best practices, and know about the available tools, such as guidelines for planning and conducting animal-based studies [25, 43]. Recently, a case study demonstrated the impact of conducting preclinical systematic reviews on the quality and transparency of research and researchers' awareness and motivation to promote change within their fields [44]. A critical comment was that many had not previously known how to report their research adequately, nor had they realized the importance of accurate reporting. Through systematic reviews, they became aware of the low reporting quality, and they became completer and more precise in the way they planned, executed, and reported their study. They also changed their view on the necessity to improve their team and research field. Hence, the assumption that this topic is well known and recognized among researchers may be wrong. There seems to be a need for more thorough education within this science field to implement rigor in one's preclinical animal study.

To accelerate progress, we conclude that educational institutions must look closer to home and support and increase educational activities of relevant teaching and training in designing and reporting animal studies. It is disappointing that large prestigious public research institutions fail to adequately report study characteristics (and also that the institutions assess publications with poor study quality at the same level as publications with high study quality). Proper education is necessary, and knowledge from education in systematic review methodology and conduction of randomized clinical control studies may guide how to approach the topic. Initiatives such as collaborative research groups and networks that serve as a backbone for this strategy should be prioritized.

## Supporting information

**S1 File. Protocol.**
(DOCX)

**S2 File. Supplementary material.**
(DOCX)

**S3 File. Agreement data.**
(XLSX)

**S4 File. PRISMA 2020 checklist.**
(DOCX)

**S1 Table. Data extraction form.**
(DOCX)

**S2 Table. Criteria for the assessment of reporting quality.**
(DOCX)

## Acknowledgments

We are grateful to Professor Malcolm Macleod, Centre for Clinical Brain Sciences, University of Edinburgh, Edinburgh, the United Kingdom for inspiring to the project, to Morten Frydenberg, Department of Public Health at Aarhus University in Denmark, for advice on random

sampling, statistical analyses and expert advice on the statistical graphs, and to Karen Rodrí-guez Sigaard, Aarhus University Library in Denmark, for aiding in the setup and modification of search strings. Dr. Alexandra Bannach-Brown from the Collaborative Approach to Meta Analyses and Review of Animal Experimental studies (CAMARADES Research Group) is acknowledged for her contributions during her time at Aarhus University.

## Author Contributions

**Conceptualization:** Birgitte S. Kousholt, Kirstine F. Præstegaard, Jennifer C. Stone, Anders Fick Thomsen, Thea Thougaard Johansen, Merel Ritskes-Hoitinga, Gregers Wegener.

**Data curation:** Birgitte S. Kousholt, Kirstine F. Præstegaard, Jennifer C. Stone, Thea Thougaard Johansen.

**Formal analysis:** Birgitte S. Kousholt, Kirstine F. Præstegaard, Jennifer C. Stone, Gregers Wegener.

**Funding acquisition:** Birgitte S. Kousholt, Merel Ritskes-Hoitinga, Gregers Wegener.

**Investigation:** Kirstine F. Præstegaard, Jennifer C. Stone, Anders Fick Thomsen, Thea Thougaard Johansen.

**Methodology:** Birgitte S. Kousholt, Kirstine F. Præstegaard, Jennifer C. Stone, Anders Fick Thomsen, Thea Thougaard Johansen, Merel Ritskes-Hoitinga, Gregers Wegener.

**Project administration:** Birgitte S. Kousholt, Kirstine F. Præstegaard, Gregers Wegener.

**Software:** Thea Thougaard Johansen.

**Supervision:** Birgitte S. Kousholt, Merel Ritskes-Hoitinga, Gregers Wegener.

**Visualization:** Birgitte S. Kousholt, Kirstine F. Præstegaard, Jennifer C. Stone, Merel Ritskes-Hoitinga, Gregers Wegener.

**Writing – original draft:** Birgitte S. Kousholt, Kirstine F. Præstegaard.

**Writing – review & editing:** Birgitte S. Kousholt, Kirstine F. Præstegaard, Jennifer C. Stone, Anders Fick Thomsen, Thea Thougaard Johansen, Merel Ritskes-Hoitinga, Gregers Wegener.

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
