## [Decision Letter · Decision Letter 0]

10 Mar 2022

PONE-D-21-20844Reporting quality in preclinical animal experimental research in 2009 and 2018: A nationwide systematic investigation

PLOS ONE

Dear Dr. Kousholt,

Thank you for submitting your manuscript to PLOS ONE. After careful consideration, we feel that it has merit but does not fully meet PLOS ONE’s publication criteria as it currently stands. Therefore, we invite you to submit a revised version of the manuscript that addresses the points raised during the review process.

I have had difficulties securing peer reviewers for your study therefore I have made an exception that my decision will be based on my own review and one other independent peer review.

As well as addressing my specific comments and those from the independent peer reviewer, I request that the raw data for individual publications is made fully available without restriction either as part of the manuscript or in a public repository.

You present an interesting study assessing the reporting of measures to reduce the risk of bias in 2009 and 2018 in a random sample of studies with at least one author affiliated with a Danish institution.

I suggest that the discussion and conclusions should reflect that these findings are specific to Danish researchers and how this compares to the findings of other similar studies sampling from other countries or without institutional restrictions. In other words, emphasize the differences and similarities between your study and previous research.

Here are some specific sentences that require clarification or edits.

Abstract

Line 20-21: “Publications from two periods with at least one affiliation to a 21 Danish university conveys any reporting progress.” Amend sentence to clarify meaning.

Line 40-42: “Knowledge on why adequate planning, execution and reporting are of importance. We suggest thorough 42 teaching in designing and reporting animal studies.” Amend sentences to clarify meaning.

Introduction

Line 49-52: “A prevalent hindrance in reproducing experiments and lack of translation to bedside is an unsatisfactory internal validity. It refers to the extent to which appropriate methodologies safeguarding against systematic errors (bias) are implemented in the design, conduct, and analysis of an experiment” Amend sentences to clarify meaning.

Line 74-75: “In this context, realization is necessary already at the planning stages to guarantee good reporting in the end.” Amend sentence to clarify meaning.

Selection of studies

Line 122: Please give details on how you defined and identified an exploratory study.

Data extraction and analysis

Line 164: “Differences between prevalence for the 2009 and 2018 studies were 164 reported with approximate 95% confidence intervals.” Do we need the word approximate? Or are you reporting the 95% confidence intervals?

Line 337: “We envisaged that an improvement in methodological reporting would be noticeable since so many journals adopt the ARRIVE guidelines.” I suggest that this sentence should be edited to read “We envisaged that an improvement in methodological reporting would be noticeable since many journals have endorsed the ARRIVE guidelines.”

Line 370: “Professor Malcolm MacLeod” Should read Professor Malcolm Macleod.

We look forward to receiving your revised manuscript.

Kind regards,

Gillian Currie

Academic Editor

PLOS ONE

Journal Requirements:

2. We note that this manuscript is a systematic review or meta-analysis; our author guidelines therefore require that you use PRISMA guidance to help improve reporting quality of this type of study. Please upload copies of the completed PRISMA checklist as Supporting Information with a file name “PRISMA checklist”.

Additional Editor Comments (if provided):

You present an interesting study assessing the reporting of measures to reduce the risk of bias in 2009 and 2018 in a random sample of studies with at least one author affiliated with a Danish institution.

I suggest that the discussion and conclusions should reflect that these findings are specific to Danish researchers and how this compares to the findings of other similar studies sampling from other countries or without institutional restrictions.

Here are some specific sentences that require clarification or edits.

Abstract

Line 20-21: “Publications from two periods with at least one affiliation to a 21 Danish university conveys any reporting progress.” Amend sentence to clarify meaning.

Line 40-42: “Knowledge on why adequate planning, execution and reporting are of importance. We suggest thorough 42 teaching in designing and reporting animal studies.” Amend sentences to clarify meaning.

Introduction

Line 49-52: “A prevalent hindrance in reproducing experiments and lack of translation to bedside is an unsatisfactory internal validity. It refers to the extent to which appropriate methodologies safeguarding against systematic errors (bias) are implemented in the design, conduct, and analysis of an experiment” Amend sentences to clarify meaning.

Line 74-75: “In this context, realization is necessary already at the planning stages to guarantee good reporting in the end.” Amend sentence to clarify meaning.

Selection of studies

Line 122: Please give details on how you defined and identified an exploratory study.

Data extraction and analysis

Line 164: “Differences between prevalence for the 2009 and 2018 studies were 164 reported with approximate 95% confidence intervals.” Do we need the word approximate? Or are you reporting the 95% confidence intervals?

Line 337: “We envisaged that an improvement in methodological reporting would be noticeable since so many journals adopt the ARRIVE guidelines.” I suggest that this sentence should be edited to read “We envisaged that an improvement in methodological reporting would be noticeable since many journals have endorsed the ARRIVE guidelines.”

Line 370: “Professor Malcolm MacLeod” Should read Professor Malcolm Macleod.

Reviewers' comments:

Reviewer's Responses to Questions

**Comments to the Author**

1. Is the manuscript technically sound, and do the data support the conclusions?

Reviewer #1: Yes

2. Has the statistical analysis been performed appropriately and rigorously? 

Reviewer #1: Yes

3. Have the authors made all data underlying the findings in their manuscript fully available?

Reviewer #1: Yes

4. Is the manuscript presented in an intelligible fashion and written in standard English?

Reviewer #1: Yes

5. Review Comments to the Author

Reviewer #1: This is an interesting study evaluating the reporting quality of preclinical studies from academic institutions in Denmark. The authors assess quality at a timepoint prior to the ARRIVE guidelines and a more recent timepoint in 2018. Overall, this is a worthwhile effort to determine if there have been improvements in reporting quality over time, focussing on measures to reduce the risk of systematic bias in animal experiments. Given the findings of other studies, the results are not surprising and indicate that reporting of key information relating to the internal validity of a study is often omitted. In fact, the findings suggest that the reporting of many quality items has not improved substantially since 2009.

General comments:

There is a need to emphasise throughout the aims and perhaps also in the discussion how this study differs from previous studies. I think one of the key strengths is that you looked in more detail at the information provided e.g. whether the randomisation method was described.

Data availability - would you be wiling to share the raw publication level data with the scores for each paper? If extracted data were available for each paper it could (1) enable future efforts to assess reporting and (2) act as training/test data for data scientists who are creating tools to automatically assess papers for risk of bias reporting.

Introduction

Line 49: Perhaps amend this sentence as it does not read well. As reproducibility and translation have just been mentioned, you could shorten to “A prevalent issue is unsatisfactory internal validity”

Line 53: Attrition is described as a safeguard here which is confusing – is there another way to phrase this?

Line 54-5: “Evidence exists that lack of reporting corresponds to the absence of conduct.” – could you add a citation to support this?

Line 57: “compared with studies not taking these precautions” – change to not reporting these precautions as we cannot be sure that researchers did not control for biases

Line 57-58: I am unclear what this means and unsure how relevant this is to the rest of the narrative

As you mention, there have been many previous studies looking at reporting quality in preclinical research. Therefore, it would be good to set out how your study differs from existing research in greater detail. Your study is focussed on specific Danish research institutions so it would be useful to explain the rationale behind this and refer to the landscape of preclinical research within Denmark.

Methods

Line 86: Was this estimation based on previous research or calculated in some way?

Line 98: I see that the search strategy is based on the SYRCLE animal filter, but could it also be shown in full in the supplementary information?

Line 101: What are the 5 Danish universities and why were they selected?

Line 105: Inconsistencies in study number descriptions (written in text for 2009, then numerical values for 2018).

These values would be better placed in the results section for clarity

For the quality of reporting items, why did you select these 10? As you mention several risk of bias tools and the ARRIVE guidelines, it is unclear how you arrived at this list and didn’t include the sex of the animals for example (from ARRIVE essential 10) but included health status. I suspect this may be due to the nature of the study capturing all different types of animal research, but it would be good to add some further information on this decision.

Results

Line 169: Repetition of prevalence being reported with 95% confidence intervals

Line 155: Was this score (0-3) used for any summary statistics or analysis?

Line 182: for random allocation, the percentage values have been calculated based on the full 250 papers rather than the subset of applicable papers which seems incorrect

I was going to suggest kappa values for reviewer agreement and see that an analysis has been performed as shown in the supplementary data file – it may be good to include a summary of this in the manuscript and highlight items which reviewers did not always agree on.

Discussion:

The discussion is well written and raises some good points. I wonder whether there could again be a mention of Denmark here and how these findings fit in to any country-specific research improvement activities and how the findings may also apply to other European institutions.

You mention here that the 5 institutions investigated are large public research institutions – I wondered whether it would be feasible to do a post-hoc analysis of study quality between these institutions, which may provide some insight into what a specific university may be doing right vs wrong.

6. PLOS authors have the option to publish the peer review history of their article (what does this mean?). If published, this will include your full peer review and any attached files.

Reviewer #1: No

---

## [Author Response · Author response to Decision Letter 0]

26 May 2022

Response to reviewers:

Dear Academic Editor, Dr. Gilllian Currie and Reviewer #1

We appreciate the efforts put into reviewing our paper (PONE-D-21-20844) and thank you for your comments. The paper has been revised accordingly. Please find our point-by-point reply below. In addition to this rebuttal letter labeled 'Response to Reviewers', we have uploaded a marked-up copy of the manuscript that highlights changes made to the original version labeled 'Revised Manuscript with Track Changes', and an unmarked version of the revised paper without tracked changes labeled 'Manuscript' as requested. 

Shortly after the upload we received an email concerning a request for a PRISMA check list. We have uploaded a checklist but do however feel that this may be a bit wrong to do so. Please find our reply to this request below in response to journal requirements item 2. 

Response to Dr Currie’s comments and requests:

As well as addressing my specific comments and those from the independent peer reviewer, I request that the raw data for individual publications is made fully available without restriction either as part of the manuscript or in a public repository.

Response: 

We agree that access to raw data is of value to the scientific community. We would however, like to work further with the data ourselves and wonder if PLOS One has any experience with delayed release of raw data? We are currently looking into the best way to make data accessible. 

You present an interesting study assessing the reporting of measures to reduce the risk of bias in 2009 and 2018 in a random sample of studies with at least one author affiliated with a Danish institution. I suggest that the discussion and conclusions should reflect that these findings are specific to Danish researchers and how this compares to the findings of other similar studies sampling from other countries or without institutional restrictions. In other words, emphasize the differences and similarities between your study and previous research.

Response: 

Thank you very much for this comment. We agree that it is important to let the discussion and conclusion reflect the study’s specific findings compared to other similar studies. We have emphasized similarities and differences in the following:

Line 72-78: “Previous research has investigated the prevalence of reporting of measures to reduce the risk of bias for specific animal disease models or subjects of interest [29-32]. Other previous evaluations of preclinical reporting have provided an overview of the reporting status of items related to the internal validity or rigor of these experiments (e.g. blinding and randomization) [33]. Taking a step further, this study investigates the reported information's level of detail by assessing preclinical studies within all animal experimental research fields with one or more authors affiliated with Danish research institutions.”.

Line 281-282: “We further researched the level of detail in the information disclosed and found the level of detail was very limited.”

Line 288-289: “Many studies additionally have very complex study designs and the precautions taken to limit bias should be sufficiently reported.”

Line 291-293: “In general, essential details related to randomization, such as the allocation method and sequence, were rarely conveyed. This fact is corroborated in studies of specific animal models of acute lung injury by Avey et al. They operationalized the ARRIVE guidelines to determine completeness in reporting and found no random sequence generation reporting [31]. Ting et al. similarly disclosed that no studies revealed the allocation method in experimental animal studies of rheumatology [29]. Our investigation concludes that there seems to be a general challenge across study fields.

Line 312-313: “A more recent study of the veterinary literature that focused on reporting adherence to the ARRIVE guidelines…”

Line 349-351: “We show that the reported information’s level of detail is generally incomplete. The incomplete reporting of these details directly impedes the ability to assess the validity of the experiments.”.

Response to specific sentences that require clarification or edits.

Abstract

Line 20-21: “Publications from two periods with at least one affiliation to a 21 Danish university conveys any reporting progress.” Amend sentence to clarify meaning.

Response:

To clarify meaning we have amended the sentence in line 19-21 to “Publications were from two time periods to convey any reporting progress and had at least one author affiliated to a Danish University.”. 

Line 40-42: “Knowledge on why adequate planning, execution and reporting are of importance. We suggest thorough 42 teaching in designing and reporting animal studies.” Amend sentences to clarify meaning.

Response:

To clarify meaning we have edited the sentence to: “We suggest thorough teaching in designing, conducting and reporting animal studies. Education in systematic review methodology should be implemented in this training and will increase motivation and behavior working towards quality improvements in science.”. in line 35-38.

Introduction

Line 49-52: “A prevalent hindrance in reproducing experiments and lack of translation to bedside is an unsatisfactory internal validity. It refers to the extent to which appropriate methodologies safeguarding against systematic errors (bias) are implemented in the design, conduct, and analysis of an experiment” Amend sentences to clarify meaning.

Response:

Thank you for noticing this. Instead of “it refers..” the sentence now includes “the internal validity”. According to reviewer one’s request, “a prevalent hindrance in reproducing experiments and lack of translation to bedside” has been changed to “..issue..”. The sentences are edited to “A prevalent issue is unsatisfactory internal validity [4]. Internal validity refers to how appropriate methodologies safeguarding against systematic errors (bias) are implemented in the design, conduct, and analysis of an experiment [5]. in line 43-45.

Line 74-75: “In this context, realization is necessary already at the planning stages to guarantee good reporting in the end.” Amend sentence to clarify meaning.

Response:

Thank you for this comment. We have clarified the meaning by rewriting the sentence: 

Line 68-71: “The implementation may be hindered by the lack of engagement of multiple stakeholders who all must engage in improving the reporting quality. In this context, researchers’ use of the guideline is necessary already in the planning stages to guarantee good reporting.”.

Materials and methods

Selection of studies

Line 122: Please give details on how you defined and identified an exploratory study.

Response: 

We defined an exploratory study as research connecting ideas to understand cause-effect and investigating novel relevant questions that have not previously been thoroughly studied. These studies do not test but generate hypotheses. 

To clarify this we have added “…(i.e. studies investigating novel questions and hypothesis-generating studies)…”. The sentence now states:

Line 119-123: “The exclusion of studies was based on the following exclusion criteria: science related to farming, wild animals or invertebrates, environment, human (clinical) studies, in vitro research, not primary papers/publications, lack of abstract or full text, exploratory studies (i.e. studies investigating novel questions and hypothesis-generating studies), and studies containing no intervention or no Danish author affiliation.”

Data extraction and analysis

Line 164: “Differences between prevalence for the 2009 and 2018 studies were 164 reported with approximate 95% confidence intervals.” Do we need the word approximate? Or are you reporting the 95% confidence intervals?

Response:

Thank you for pointing this out. The word approximate is not necessary and have been omitted. 

Line 167-168: “Prevalence and differences between prevalence for the 2009 and 2018 studies were reported with 95% confidence intervals.” 

Line 337: “We envisaged that an improvement in methodological reporting would be noticeable since so many journals adopt the ARRIVE guidelines.” I suggest that this sentence should be edited to read “We envisaged that an improvement in methodological reporting would be noticeable since many journals have endorsed the ARRIVE guidelines.”

Response:

Thank you. The sentence is corrected according to your request.

Line 347-348: “We envisaged that an improvement in methodological reporting would be noticeable since many journals have endorsed the ARRIVE guidelines.” 

Line 370: “Professor Malcolm MacLeod” Should read Professor Malcolm Macleod.

Response: 

Thank you for pointing out this misspelling. This is now corrected to read “Professor Malcolm Macleod” in line 382.

Response to journal requirements:

Response:

We have revisited PLOS ONE’s style requirements as well as followed the templates for correcting the formatting:

Line 14: “BSK” is added to the corresponding author email

Line 16, 22, 29, 37: We have deleted the subtitles “objective”, “methods”, “results, and “conclusion” from the abstract.

Line 128-129: Figure title is written in bold type

Line 179: “table 1” is corrected to “Table 1”

Line 262: Table 1 title is written in bold type, 12 pt

Line 263-264: table and legend are written in 12 pt

In addition, supporting information files have been renamed to match the supporting information captions within the manuscript.

2. We note that this manuscript is a systematic review or meta-analysis; our author guidelines therefore require that you use PRISMA guidance to help improve reporting quality of this type of study. Please upload copies of the completed PRISMA checklist as Supporting Information with a file name “PRISMA checklist”.

Response:

Thank you for this comment. It leads to some confusion. We do not perceive the manuscript as a systematic review per se and several items from the PRISMA checklist is not applicable. This is due to the fact that no summary of all individual studies over a specific health-related issue is given. Furthermore, the study does not inform decisions about the safety and efficacy of a treatment for participants in clinical trials but rather convey the reporting quality and progress in reporting in preclinical studies. We do however agree that part of the systematic review methodology is applied and this leads to the results presented in this investigation. 

Early on in the submission process we were in contact with PLOS ONE and received a reply in august 2021 confirming that this view was accepted. If you have changed your view on this please let us now for further discussion. We have for the sake of moving the manuscript further uploaded the checklist with file name “S4_file.word”. In the manuscript we have added “S4 File. PRISMA checklist” in line 500. 

Response:

The following two references have been added to the reference list (reference number 8 and 9 in line 52 in the manuscript)

Line 406-407: Tikka, C., et al., Quality of reporting and risk of bias: a review of randomised trials in occupational health. Occupational and Environmental Medicine, 2021. 78(9): p. 691-696.

Line 408-409 :Riley, S.P., et al., A systematic review of orthopaedic manual therapy randomized clinical trials quality. The Journal of manual & manipulative therapy, 2016. 24(5): p. 241-252 

The following two references have been edited:

Line 481-482: The reference “Hair, K., M.R. Macleod, and E.S. Sena, A randomised controlled trial of an Intervention to Improve Compliance with the ARRIVE guidelines (IICARus). bioRxiv, 2018” is now published in Research Integrity and Peer Review doi: 10.1186/s41073-019-0069-3. Thus the reference has been updated to “Hair, K., et al., A randomised controlled trial of an Intervention to Improve Compliance with the ARRIVE guidelines (IICARus). Research Integrity and Peer Review, 2019. 4(1): p. 12.”

Line 492-494: The reference number 45 “Z.s. Modules - Programme "More knowledge with fewer animals". 2020; Available from: https://www.zonmw.nl/nl/actueel/nieuws/detail/item/systematisch-literatuuronderzoek-vervangt-vermindert-en-verfijnt-proefdieronderzoek/.” is replaced since data from this case study has been published in 2021. The reference number 45 in line 363 in the manuscript is corrected to: Menon, J.M.L., et al., The impact of conducting preclinical systematic reviews on researchers and their research: A mixed method case study. PloS one, 2021. 16(12): p. e0260619-e0260619.

Reviewers' comments:

Reviewer #1: This is an interesting study evaluating the reporting quality of preclinical studies from academic institutions in Denmark. The authors assess quality at a timepoint prior to the ARRIVE guidelines and a more recent time point in 2018. Overall, this is a worthwhile effort to determine if there have been improvements in reporting quality over time, focusing on measures to reduce the risk of systematic bias in animal experiments. Given the findings of other studies, the results are not surprising and indicate that reporting of key information relating to the internal validity of a study is often omitted. In fact, the findings suggest that the reporting of many quality items has not improved substantially since 2009. 

Response to general comments.

There is a need to emphasise throughout the aims and perhaps also in the discussion how this study differs from previous studies. I think one of the key strengths is that you looked in more detail at the information provided e.g. whether the randomisation method was described.

Response:

Thank you for this comment. We have responded to this matter earlier on in this letter. Please refer to the earlier response beginning with: “Thank you very much for this comment. We agree that it is important to let the discussion and conclusion reflect the study’s specific findings compared to other similar studies. We have emphasized similarities and differences in the following…” p. 2 in this letter.

Data availability - would you be willing to share the raw publication level data with the scores for each paper? If extracted data were available for each paper it could (1) enable future efforts to assess reporting and (2) act as training/test data for data scientists who are creating tools to automatically assess papers for risk of bias reporting.

Response: 

We agree that access to raw data is of value to the scientific community. We would however, like to work further with the data ourselves and wonder if PLOS One has any experience with delayed release of raw data? We are currently looking into the best way to make data accessible. 

Introduction

Line 49: Perhaps amend this sentence as it does not read well. As reproducibility and translation have just been mentioned, you could shorten to “A prevalent issue is unsatisfactory internal validity”

Response: 

The sentence is edited to “A prevalent issue is unsatisfactory internal validity” in line 43.

Line 53: Attrition is described as a safeguard here which is confusing – is there another way to phrase this?

Response:

Attrition is now rephrased to “description of animals’ and samples’ flow including reasons for exclusion

Line 46-47 is edited to “Essential safeguards are blinding, randomization, and a thorough description of animals’ and samples’ flow including reasons for exclusion [6].”. 

Line 54-5: “Evidence exists that lack of reporting corresponds to the absence of conduct.” – could you add a citation to support this?

Response:

The main issue with poor reporting is that risk of bias becomes difficult to assess. So this usually leads to unclear (or some concerns) risk of bias judgments for the overall study. We have added two references from clinical trials “[8, 9]” in line 49 showing that poorly reported trials had a higher risk of bias. 

Line 57: “compared with studies not taking these precautions” – change to not reporting these precautions as we cannot be sure that researchers did not control for biases

Response:

The word “taking” has been corrected to “reporting” in line 51.

Line 57-58: I am unclear what this means and unsure how relevant this is to the rest of the narrative.

As you mention, there have been many previous studies looking at reporting quality in preclinical research. Therefore, it would be good to set out how your study differs from existing research in greater detail. Your study is focused on specific Danish research institutions so it would be useful to explain the rationale behind this and refer to the landscape of preclinical research within Denmark.

Response: 

Thank you for these comments. Regarding the meta-epidemiological studies from clinical data, we regard them as important since they corroborate findings in preclinical research and furthermore further researches the additive effect of more than one bias. We anticipate that this is equivalent in preclinical research.

Line 51-53: “This finding is corroborated in meta-epidemiological studies of clinical data that identify a negative additive impact when more than one safeguard is omitted [14-17].”. 

In terms of how our study differs from existing research we have emphasized this in the following:

Line 72-78: “Previous research has investigated the prevalence of reporting of measures to reduce the risk of bias for specific animal disease models or subjects of interest [29-32]. Other previous evaluations of preclinical reporting have provided an overview of the reporting status of items related to the internal validity or rigor of these experiments (e.g. blinding and randomization) [33]. Taking a step further, this study investigates the reported information's level of detail by assessing preclinical studies within all animal experimental research fields with one or more authors affiliated with Danish research institutions.”.

Line 281-282: “We further researched the level of detail in the information disclosed and found the level of detail was very limited.”

Line 288-289: “Many studies additionally have very complex study designs and the precautions taken to limit bias should be sufficiently reported.”

Line 291-297: “In general, essential details related to randomization, such as the allocation method and sequence, were rarely conveyed. This fact is corroborated in studies of specific animal models of acute lung injury by Avey et al. They operationalized the ARRIVE guidelines to determine completeness in reporting and found no random sequence generation reporting [31]. Ting et al. similarly disclosed that no studies revealed the allocation method in experimental animal studies of rheumatology [29]. Our investigation concludes that there seems to be a general challenge across study fields.

Line 312-313: “A more recent study of the veterinary literature that focused on reporting adherence to the ARRIVE guidelines…”

Line 347-351: “We envisaged that an improvement in methodological reporting would be noticeable since many journals have endorsed the ARRIVE guidelines. However, advancements continue to progress at a slow pace or do not happen at all. We show that the reported information’s level of detail is generally incomplete. The incomplete reporting of these details directly impedes the ability to assess the validity of the experiments.”.

In regard to selection of specific Danish research institutions: These are the leading research institutions in Denmark reporting animal experimental research and of interest for this study.

Methods

Line 86: Was this estimation based on previous research or calculated in some way?

Response: 

The estimation is based on pilot data from an internal study of published animal research from Aarhus University but primarily on group discussion taking into account the amount of time available for thorough and in-depth assessment of publications. It was estimated that 500 papers in total could be performed. Retrospectively, because we used simple random sampling, we assume that the remaining 1251 studies (3051-1800) contain 370 eligible studies (30 %). This assumption is based on the fact that of 1800 studies (identified by random sampling) screened for eligibility, we retrieved 531 eligible studies equalling 30%.

The sentence in line 86-88 now states: “It was estimated that a thorough assessment of 500 papers in total – 250 papers from each year – could be performed within the given timeframe.”.

Line 98: I see that the search strategy is based on the SYRCLE animal filter, but could it also be shown in full in the supplementary information?

Response: 

Thank you for this comment. The syntax for the SYRCLE animal filter has now been included as a supporting information file (S1 File) and the search strategy is based on the search strategy found in reference number 35 and 36 in the manuscript. 

Line 101: What are the 5 Danish universities and why were they selected?

Response:

This study include papers reporting animal experiments from Aarhus University (AU), Aalborg University (AAU), University of Copenhagen (UCPH), University of Southern Denmark (SDU) or Technical University of Denmark (DTU). This information is found in the supporting information S1 File. These are the leading research institutions in Denmark reporting animal experimental research and thus of most interest for this study. 

Line 105: Inconsistencies in study number descriptions (written in text for 2009, then numerical values for 2018). These values would be better placed in the results section for clarity.

Response:

Thank you for noticing this. The sentence in line 105-107 is amended to: “One thousand, one hundred and sixty-one studies from 2009 and 1890 studies from 2018 were found.”.

In our opinion this is not part of the results but is part of the method used to reach the 250 papers from each year. It is part of figure 1 and we feel that it is very important to keep the figure in relation to the methods section for readers to more easily understand the flow. We have now mentioned this in the results section as well; in line 173: “The flow of the publications retrieved is described in the Methods section and in figure 1.”.

For the quality of reporting items, why did you select these 10? As you mention several risk of bias tools and the ARRIVE guidelines, it is unclear how you arrived at this list and didn’t include the sex of the animals for example (from ARRIVE essential 10) but included health status. I suspect this may be due to the nature of the study capturing all different types of animal research, but it would be good to add some further information on this decision.

Response:

Thank you for this comment. We chose the Landis 4 criteria and a number of representative items of the ARRIVE guidelines. We found it more important in this study to investigate the level of detail and thus we ended up with 10 items in total. The Landis 4 is mentioned in the ARRIVE guideline. From our veterinary background we know that health status is a really important aspect of research which role is often neglected. The animals’ health or immune status can influence their physiology and behaviour as well as their response to treatments, and thus impact on experimental variability and scientific outcomes. However, the animals' health status or comorbidities before and during experiments is to our knowledge rarely considered, and therefore we thought that this would be interesting to investigate. 

We have amended the sentence in line 133-138 to the following: “Each publication was assessed according to 10 items primarily based on the Landis four related to the quality of reporting of significant methodology and included in the ARRIVE guidelines [6, 25, 26]. The selection of items was due to the nature of the study capturing different types of animal research. One item “health status”, was chosen since it, to our knowledge, is scarcely investigated even though it may influence many research outcomes [22].”.

Results

Line 169: Repetition of prevalence being reported with 95% confidence intervals

Response: 

Thank you for noticing. The sentence in line 169 “All prevalences are reported with 95% confidence intervals.” is deleted.

Line 155: Was this score (0-3) used for any summary statistics or analysis?

Response:

We did not use this data for summary statistics. We used the information to describe the information retrieved in a descriptive manner. 

Line 182: for random allocation, the percentage values have been calculated based on the full 250 papers rather than the subset of applicable papers which seems incorrect

Response: 

Thank you for noticing this. To make a fair judgement it was necessary to find out whether the studies could be randomized or not. However, studies not reporting randomization were quite challenging as it was often difficult to find out whether they could in fact be randomized or not due to unclear study design or groups. We therefore decided on the following:

a. Studies reporting randomization (randomized) or reporting that the study is non-randomized scored "yes, reported" 

b. Studies not reporting randomization, however, randomization is possible score "no, not reported"

c. Studies not reporting randomization as randomization is not possible (non-randomized). We decided to score those studies “no, not reported” if nothing was reported about randomization, and write a comment that randomization is not possible for that specific study. One might argue that this is not completely fair. However, in our opinion these studies ought to report that randomization was not feasible for complete reporting and transparency. 

We have discussed this in line 284-288: “In studies where blinding and randomization were not feasible, the reason why (e.g., study design) was rarely justified nor considered a limitation and acknowledged in the study report. A description of why such a precaution is not taken will bring the reader’s attention to the missing safeguard so the results can be judged accordingly”. 

I was going to suggest kappa values for reviewer agreement and see that an analysis has been performed as shown in the supplementary data file – it may be good to include a summary of this in the manuscript and highlight items which reviewers did not always agree on.

Response:

Thank you for your comment. We have calculated percent agreement and kappa as both have strengths and limitations, and as suggested in McHugh, M. L. (2012). Interrater reliability: the kappa statistic. Biochemia medica, 22(3), 276-282. We have kept our data in the supporting information S3 File and in addition commented on items reviewers did not always agree on. 

In line 169-170 we have added the sentence: “Reviewer agreement and Cohen’s Kappa values are disclosed in the supporting information (S3 File).”.

Discussion

The discussion is well written and raises some good points. I wonder whether there could again be a mention of Denmark here and how these findings fit in to any country-specific research improvement activities and how the findings may also apply to other European institutions.

You mention here that the 5 institutions investigated are large public research institutions – I wondered whether it would be feasible to do a post-hoc analysis of study quality between these institutions, which may provide some insight into what a specific university may be doing right vs wrong.

Response:

Thank you very much for this suggestion. Comparison of the current reporting status between five Danish universities was originally a part of our study and can be found in the supporting information S1 File. However, this was not feasible due to the experimental design and the procedure for random sampling, and we decided to look at this on a national level. 

In addition, the following has been amended:

Line 20-21: The following sentence: “Using a predefined research protocol and a systematic search, we retrieved all relevant animal studies.” is edited to “We retrieved all relevant animal experimental studies using a predefined research protocol and a systematic search.”.

Line 30: “the method of random allocation” is edited to “random allocation method”

Line 61: “…fore example” is corrected to “for example”

Line 80: “…each of the reported items.” is amended to “...each reported item.”

Line 85-86 is corrected to “An equal number of studies from each year were included to compare the results between the two time periods.”

Line 87: “…and in depth…” is deleted.

Line 101: “…to…” is corrected to “…with…”

Line 110-112: “Due to the decision to perform a comprehensive search strategy to identify all relevant preclinical animal studies, the majority of the studies were not applicable.”. 

Line 119: “Exclusion…” is corrected to “The exclusion…”

Line 132-133: “…to allow the assessment of reporting…” is amended to “and to assess reporting…”

Line 142: “…in addition to…” is corrected to “…and…”

Line 149 is amended to “…complete information defined in the item or where items were reported not conducted…”.

Line 152: “Annotations and quotes for each item…” is edited to “Each item’s annotations and quotes…”. 

Line 154: “After completing the initial reporting quality assessments by each reviewer…” is edited to “After completing each reviewer’s initial reporting quality assessments…”. 

Line 167: “table” is corrected to “tables”

Line 178: “presented” is corrected to “shown”

Line 194 and 196: “…the…” is added.

Line 224: “…that…” is deleted.

Line 232: “…the…” is added.

Line 247: “reason” is corrected to “reasons”.

Line 248: “…exact…” is corrected to “…precise…”.

Line 248: “reason” is corrected to “reasons”.

Line 259: “…to be present…” is deleted.

Line 262: Table 1 title is amended to “Table 1. Prevalence of reporting details in Danish preclinical research in 2009 compared to 2018.”.

Line 285: “…why…” is deleted.

Line 299: “Interestingly, successful randomization may be negatively influenced by small sample sizes…” is amended to “Interestingly, small sample sizes may negatively influence successful randomization…”. 

Line 311: “rather” is deleted.

Line 317: “…then…” is deleted.

Line 318: “In general…” is deleted.

Line 328: “We found that several…” is amended to “Several…”.

Line 332: “…with…” is added.

Line 342-344: The sentence: “A fully disclosed health report based on a case-oriented approach to the FELASA (Federation of European Laboratory Animal Science Associations) guidelines should be mandatory [23, 24]. Has been amended to “A fully disclosed health report should be mandatory and based on a case-oriented approach to the FELASA (Federation of European Laboratory Animal Science Associations) guidelines [23, 24].”.

Line 351: “is unable to” is corrected to “cannot”.

Line 377: “be used as guidance on…” is corrected to “guide”. 

Line 383: “…the…” is added.

We are looking forward to your reply.

On behalf of the authors.

Kind regards,

Dr. Kousholt

---

## [Editor Report · Decision Letter 1]

28 Jul 2022

PONE-D-21-20844R1Reporting quality in preclinical animal experimental research in 2009 and 2018: A nationwide systematic investigationPLOS ONE

Dear Dr. Kousholt,

Thank you for submitting your manuscript to PLOS ONE. After careful consideration, we feel that it has merit but does not fully meet PLOS ONE’s publication criteria as it currently stands. Therefore, we invite you to submit a revised version of the manuscript that addresses the points raised during the review process.

Thank you for your efforts revising your manuscript in response to our suggestions.

My main concern is that your data should be made available- this is a requirement for publication. PLOS journals require authors to make all data necessary to replicate their study’s findings publicly available without restriction at the time of publication. The specific details can be found here: https://journals.plos.org/plosone/s/data-availability#loc-unacceptable-data-access-restrictions

I do not believe that a PRISMA checklist is necessary for this study but it is useful to include the graphical representation of the flow of studies (Figure 1).

I have made some further suggestions to changes to the text to help clarify meaning.

In addition, it is important to clarify how the reviewers were able to identify exploratory studies. 

We look forward to receiving your revised manuscript.

Kind regards,

Gillian Currie

Academic Editor

PLOS ONE

Journal Requirements:

Additional Editor Comments:

Thank you for your efforts revising your manuscript in response to our suggestions.

My main concern is that your data should be made available- this is a requirement for publication. PLOS journals require authors to make all data necessary to replicate their study’s findings publicly available without restriction at the time of publication. The specific details can be found here: https://journals.plos.org/plosone/s/data-availability#loc-unacceptable-data-access-restrictions

I do not believe that a PRISMA checklist is necessary for this study but it is useful to include the graphical representation of the flow of studies (Figure 1).

Further suggestions to your specific text changes, to clarify meaning, are below:

1. Line 72-78: “Previous research has investigated the prevalence of reporting of measures to reduce the risk of bias for specific animal disease models or subjects of interest [29-32]. Other previous evaluations of preclinical reporting have provided an overview of the reporting status of items related to the internal validity or rigor of these experiments (e.g. blinding and randomization) [33]. Taking a step further, this study investigates the reported information's level of detail by assessing preclinical studies within all animal experimental research fields with one or more authors affiliated with Danish research institutions.”.

I suggest that you remove the “Taking a step further” and instead “This study investigates the reported information's level of detail by assessing preclinical studies within all animal experimental research fields with one or more authors affiliated with Danish research institutions.”.

2. Line 49-52: “A prevalent hindrance in reproducing experiments and lack of translation to bedside is an unsatisfactory internal validity. It refers to the extent to which appropriate methodologies safeguarding against systematic errors (bias) are implemented in the design, conduct, and analysis of an experiment” Amend sentences to clarify meaning.

Response:

Thank you for noticing this. Instead of “it refers..” the sentence now includes “the internal validity”. According to reviewer one’s request, “a prevalent hindrance in reproducing experiments and lack of translation to bedside” has been changed to “..issue..”. The sentences are edited to “A prevalent issue is unsatisfactory internal validity [4]. Internal validity refers to how appropriate methodologies safeguarding against systematic errors (bias) are implemented in the design, conduct, and analysis of an experiment [5]. in line 43-45.

I do not think that the sentence “Internal validity refers to how appropriate methodologies safeguarding against systematic errors (bias) are implemented in the design, conduct, and analysis of an experiment” is accurate (this is not a definition of internal validity that I am familiar with). Amend to clarify e.g. define internal validity and then state that methodologies to safeguard against bias can be implemented to increase internal validity.

3. Line 74-75: “In this context, realization is necessary already at the planning stages to guarantee good reporting in the end.” Amend sentence to clarify meaning.

Response:

Thank you for this comment. We have clarified the meaning by rewriting the sentence:

Line 68-71: “The implementation may be hindered by the lack of engagement of multiple stakeholders who all must engage in improving the reporting quality. In this context, researchers’ use of the guideline is necessary already in the planning stages to guarantee good reporting.”.

I suggest that the wording is changed to clarify meaning. If I have understood your meaning correctly then I suggest the wording “In this context, the use of the ARRIVE guidelines by researchers is necessary at the planning stage to help improve experimental design and, in turn, improve reporting.”

4. Line 122: Please give details on how you defined and identified an exploratory study.

Response:

We defined an exploratory study as research connecting ideas to understand cause-effect and investigating novel relevant questions that have not previously been thoroughly studied. These studies do not test but generate hypotheses.

To clarify this we have added “…(i.e. studies investigating novel questions and hypothesis-generating studies)…”. The sentence now states:

Line 119-123: “The exclusion of studies was based on the following exclusion criteria: science related to farming, wild animals or invertebrates, environment, human (clinical) studies, in vitro research, not primary papers/publications, lack of abstract or full text, exploratory studies (i.e. studies investigating novel questions and hypothesis-generating studies), and studies containing no intervention or no Danish author affiliation.”

I am still not clear as to how an exploratory study was identified. How did you decide it was a novel question? Or hypothesis-generating? Did the authors have to state this in their manuscript?
---

## [Author Response · Author response to Decision Letter 1]

26 Aug 2022

Response to reviewers:

Dear Academic Editor, Dr. Gilllian Currie

Thank you for your continued interest in our paper (PONE-D-21-20844R1) and the effort that you have put into reviewing the revised manuscript. Please find our point-by-point reply below. In addition to this rebuttal letter labeled 'Response to Reviewers', we have uploaded a marked-up copy of the manuscript that highlights changes made to the original version labeled 'Revised Manuscript with Track Changes', and an unmarked version of the revised paper without tracked changes labeled 'Manuscript' as requested. 

Response to Dr Currie’s comments and requests:

Thank you for your efforts revising your manuscript in response to our suggestions. My main concern is that your data should be made available- this is a requirement for publication. PLOS journals require authors to make all data necessary to replicate their study’s findings publicly available without restriction at the time of publication. The specific details can be found here: https://journals.plos.org/plosone/s/data-availability#loc-unacceptabledata-access-restrictions

Response:

Thank you for responding to our request. Time has passed and at this time point we are fully able to make all data necessary to replicate our study’s findings available. So it is no longer an issue for us to keep the data restricted. The datasets are currently stored in The Open Science Framework repository https://osf.io and thus ready to be made publicly available upon PLOS One’s request. Additionally, the Supporting information file, S3 File is renamed “S3 File: Agreement data” in line 501 and now only contains our agreement data. 

I do not believe that a PRISMA checklist is necessary for this study but it is useful to include the graphical representation of the flow of studies (Figure 1). 

Response:

Thank you for agreeing on this matter. We omitted the PRISMA checklist from the supporting information and maintained the graphical presentation of the flow of studies (Figure 1) in the manuscript and submitted the manuscript. We were then notified to include the checklist in order for the submission process to proceed and swiftly responded your decision in the “Enter comments” prior to re-submitting our manuscript. However, omitting the checklist resulted in a new “revisions sent back to author” notification with a request for the checklist. Thus, to resolve this and speed up the review process (currently > 1 year), we have included the PRISMA checklist and stamped it with “Not to be included in the manuscript”.

1. Line 72-78: “Previous research has investigated the prevalence of reporting of measures to reduce the risk of bias for specific animal disease models or subjects of interest [29-32]. Other previous evaluations of preclinical reporting have provided an overview of the reporting status of items related to the internal validity or rigor of these experiments (e.g. blinding and randomization) [33]. Taking a step further, this study investigates the reported information's level of detail by assessing preclinical studies within all animal experimental research fields with one or more authors affiliated with Danish research institutions.” I suggest that you remove the “Taking a step further” and instead “This study investigates the reported information's level of detail by assessing preclinical studies within all animal experimental research fields with one or more authors affiliated with Danish research institutions.”. 

Response: 

Thank you for this comment. The sentence is corrected according to your request. You will find the sentence: “This study investigates the reported information's level of detail by assessing preclinical studies within all animal experimental research fields with one or more authors affiliated with Danish research institutions.” in line 75-78.

2. Line 49-52: 

I do not think that the sentence “Internal validity refers to how appropriate methodologies safeguarding against systematic errors (bias) are implemented in the design, conduct, and analysis of an experiment” is accurate (this is not a definition of internal validity that I am familiar with). Amend to clarify e.g. define internal validity and then state that methodologies to safeguard against bias can be implemented to increase internal validity. 

Response: 

We agree that the sentences indeed need an additional revision in order to be clear and concise and thank you for underscoring this fact. We suggest the following: 

Line 43-46: “A prevalent issue is unsatisfactory internal validity. Internal validity is the extent to which a design and conduct of a study eliminates the possibility of systematic errors (bias) [4]. Appropriate methodologies safeguarding against systematic errors can be implemented in the design, conduct, and analysis of an experiment in order to increase the internal validity [4].”

3. Line 74-75: “In this context, realization is necessary already at the planning stages to guarantee good reporting in the end.” Amend sentence to clarify meaning. Response: Thank you for this comment. We have clarified the meaning by rewriting the sentence: Line 68-71: “The implementation may be hindered by the lack of engagement of multiple stakeholders who all must engage in improving the reporting quality. In this context, researchers’ use of the guideline is necessary already in the planning stages to guarantee good reporting.”. I suggest that the wording is changed to clarify meaning. If I have understood your meaning correctly then I suggest the wording “In this context, the use of the ARRIVE guidelines by researchers is necessary at the planning stage to help improve experimental design and, in turn, improve reporting.” 

Response:

Thank you for this comment. We have clarified the meaning of the sentence by revising according to your suggestion: “The implementation may be hindered by the lack of engagement of multiple stakeholders who all must engage in improving the reporting quality. In this context, the use of the ARRIVE guideline by researchers is necessary already at the planning stage to help improve experimental design and, in turn, improve reporting.” in line 67-71. 

4. Line 122: Please give details on how you defined and identified an exploratory study. Response: We defined an exploratory study as research connecting ideas to understand cause-effect and investigating novel relevant questions that have not previously been thoroughly studied. These studies do not test but generate hypotheses. To clarify this we have added “…(i.e. studies investigating novel questions and hypothesis-generating studies)…”. The sentence now states: Line 119-123: “The exclusion of studies was based on the following exclusion criteria: science related to farming, wild animals or invertebrates, environment, human (clinical) studies, in vitro research, not primary papers/publications, lack of abstract or full text, exploratory studies (i.e. studies investigating novel questions and hypothesis-generating studies), and studies containing no intervention or no Danish author affiliation.” I am still not clear as to how an exploratory study was identified. How did you decide it was a novel question? Or hypothesis-generating? Did the authors have to state this in their manuscript? 

Response: 

Thank you for your comment. Exploratory studies were identified through study author statements that the study was explorative, or the study was assessed to be hypothesis-generating and performed to e. g. investigate novel questions, develop models or investigate and validate new animal models, look for biomarkers, explore pharmacokinetics/biodistribution, gene identification/expression, or mapping of receptors or microcirculation”.

To clarify, we have amended the sentence in line 119-125: “The exclusion of studies was based on the following exclusion criteria: science related to farming, wild animals or invertebrates, environment, human (clinical) studies, in vitro research, not primary papers/publications, lack of abstract or full text, studies containing no intervention or no Danish author affiliation, and exploratory studies (the latter studies were identified through study author statements that the study was explorative, or studies were assessed to investigate novel questions and to be hypothesis-generating).”

We are looking forward to your reply.

On behalf of the authors. 

Kind regards,

Dr. Kousholt

---

## [Editor Report · Decision Letter 2]

27 Sep 2022

Reporting quality in preclinical animal experimental research in 2009 and 2018: A nationwide systematic investigation

PONE-D-21-20844R2

Dear Dr. Kousholt,

We’re pleased to inform you that your manuscript has been judged scientifically suitable for publication and will be formally accepted for publication once it meets all outstanding technical requirements.

Kind regards,

Gillian Currie

Academic Editor

PLOS ONE
---

## [Editor Report · Acceptance letter]

12 Oct 2022

PONE-D-21-20844R2 

Reporting quality in preclinical animal experimental research in 2009 and 2018: A nationwide systematic investigation 

Dear Dr. Kousholt:

I'm pleased to inform you that your manuscript has been deemed suitable for publication in PLOS ONE. Congratulations! Your manuscript is now with our production department. 

Kind regards, 

on behalf of

Dr. Gillian Currie 

Academic Editor

PLOS ONE